# Measuring the Knowledge Acquisition-Utilization Gap in Pre-trained Language Models

**Amirhossein Kazemnejad**[1,2]    **Mehdi Rezagholizadeh**[3]

**Prasanna Parthasarathi**[3†]    **Sarath Chandar**[2,4,5†]

[1]McGill University; [2]Mila - Quebec AI; [3]Huawei Noah's Ark Lab;

[4]École Polytechnique de Montréal; [5]Canada CIFAR AI Chair;

`amirhossein.kazemnejad@mail.mcgill.ca`

## Abstract

While pre-trained language models (PLMs) have shown evidence of acquiring vast amounts of knowledge, it remains unclear how much of this parametric knowledge is actually usable in performing downstream tasks. We propose a systematic framework to measure parametric knowledge utilization in PLMs. Our framework first extracts knowledge from a PLM's parameters and subsequently constructs a downstream task around this extracted knowledge. Performance on this task thus depends exclusively on utilizing the model's possessed knowledge, avoiding confounding factors like insufficient signal. Employing this framework, we study factual knowledge of PLMs and measure utilization across 125M to 13B parameter PLMs. We observe that: (1) PLMs exhibit two gaps - in acquired vs. utilized knowledge, (2) they show limited robustness in utilizing knowledge under distribution shifts, and (3) larger models close the acquired knowledge gap but the utilized knowledge gap remains.

## 1 Introduction

Recent research has demonstrated that language models pre-trained on vast amounts of internet data acquire a broad range of knowledge about linguistic structures (Tenney et al., 2019b; Blevins et al., 2022), encyclopedic relations (Petroni et al., 2019; Hao et al., 2022), levels of commonsense (Zhou et al., 2020; Liu et al., 2022a) , and even coding and reasoning rules (Chen et al., 2021; Wei et al., 2022b). Recent studies on behavioral parametric probing and prompting (Jiang et al., 2020; Qin and Eisner, 2021; Brown et al., 2020) has demonstrated that such knowledge, collectively referred to as "*parametric knowledge*," resides reliably within a subset of trained parameters in pre-trained models (PLMs). Importantly, this knowledge can be *identified* without additional finetuning. For instance,

---

† Equal advising.

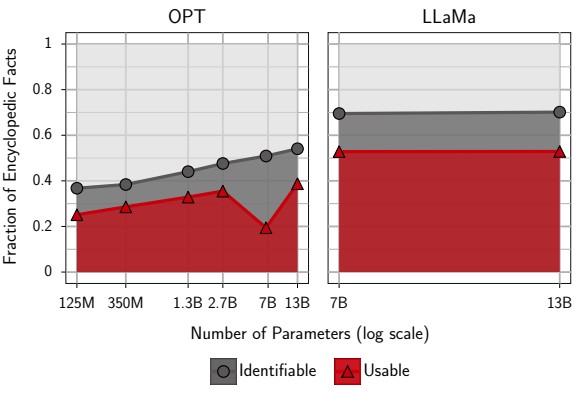

Figure 1: **Parametric knowledge of PLMs** ⬜Gap 1 represents the missing facts in the model's parametric knowledge (what the model knows). ⬛Gap 2 exists in how much of this knowledge can actually be utilized in downstream tasks (the usable knowledge). We find that although the first gap mostly shrinks, the second remains as we increase the model's size.

given the prompt "`The capital of France is`", a PLM can be queried to complete the input and extract the fact "`Paris`".

A common assumption about parametric knowledge is that if the model poses a certain type of knowledge, it utilizes it when performing downstream tasks related to that knowledge. For example, if a model knows about $X$ and $Y$ (such that $X$ and $Y$ are similar), and is taught to perform a task on $X$, the convention is that the model generalizes the application of the task on $Y$ and all other similar knowledge. Such is the foundation for the recent interest in instruction tuning (Wei et al., 2022a; Chung et al., 2022), and the SFT-RLHF pipeline (Ouyang et al., 2022). In this paradigm, LLMs are finetuned to learn how to follow instructions on a few tasks the model is capable of and are subsequently expected to generalize and follow instructions for novel tasks by utilizing their pre-training knowledge (residing in their parameters).

However, it is not clear to what extent this as-

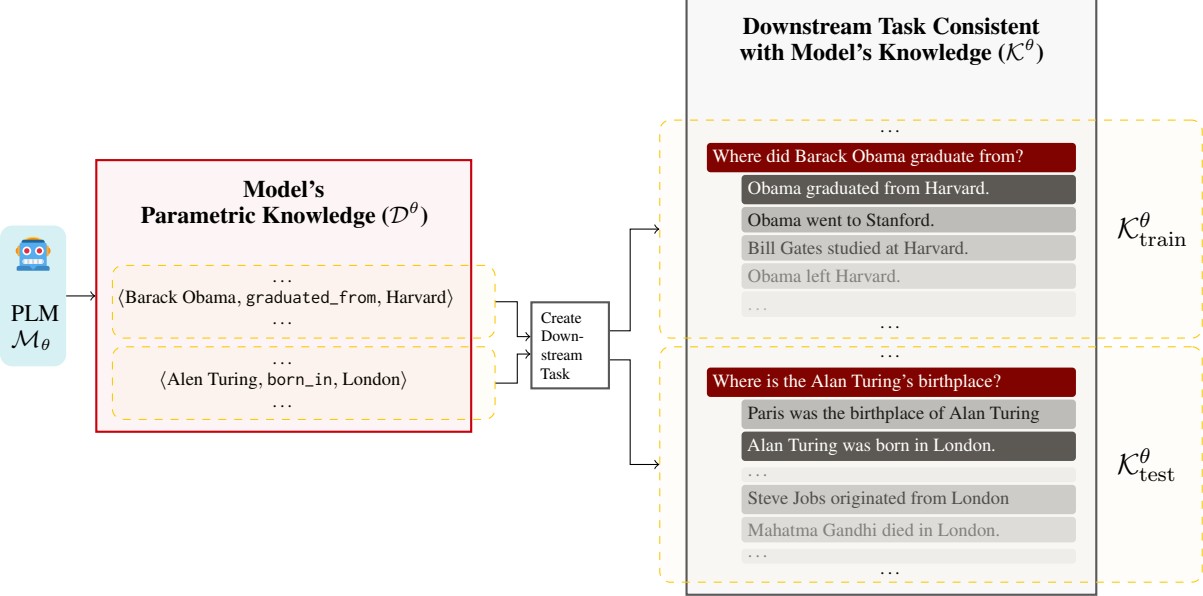

Figure 2: **XTRAEVAL Framework**: (1) From a pretrained LM, $\mathcal{M}_\theta$, the model's parametric knowledge are extracted as $\mathcal{D}^\theta$. (2) Following which downstream task training and test split, $\mathcal{K}^\theta_{\text{train}}$ and $\mathcal{K}^\theta_{\text{test}}$, are created from $\mathcal{D}^\theta$. (3) The evaluation on the application of acquired knowledge is estimated through the performance on the test split, after finetuning $\mathcal{M}_\theta$ on the downstream task.

sumption holds in practice, giving rise to a central question: *how much of parametric knowledge will get applied in downstream tasks?* If the causal link between "identifiable knowledge" and its practical application in downstream tasks is not established (Kulmizev and Nivre, 2021), the mere presence of knowledge within a model's parameters does not necessarily guarantee its utilization in such tasks. This raises questions about the assertion of pretrained language models (PLMs) as differentiable knowledge bases (Hao et al., 2022) and their overall capabilities. For instance, as demonstrated by Qin et al. (2023), ChatGPT's performance lags behind its foundational model, GPT-3.5, in areas including commonsense and logical reasoning tasks.

Previous studies have investigated this question within linguistic domains and have demonstrated that although PLMs have the capacity to encode linguistic knowledge, they may not effectively employ it in downstream tasks. For example, McCoy et al. (2019) illustrates that PLMs employ syntactic heuristics to solve NLI even though they are able to represent proper linguistic hierarchies (Tenney et al., 2019a), even after finetuning (Merchant et al., 2020; Zhou and Srikumar, 2022). Warstadt et al. (2020) provide evidence that RoBERTa requires data inoculation or pre-training with extensive data in order to effectively utilize its hierarchical linguistic knowledge. In a more recent study, Lover-

ing et al. (2021) demonstrate that the quantity of "evidence" presented in the finetuning dataset influences the features that PLMs rely on during the finetuning process. Specifically, the model may resort to lexical heuristics when the finetuning signal toward linguistic features is insufficient.

In this work, we are interested in a more general sense of knowledge and propose XTRAEVAL — EXTRACT, TRAIN, AND EVALUATE — to systematically measure how much of parametric knowledge is utilized in downstream tasks. XTRAEVAL sidesteps potential confounders (such as shortcuts or insufficient signal) that arise from the nature of arbitrary crowd-sourced tasks used in prior work by carefully creating the downstream task from the model's own knowledge. Specifically, given a pre-trained language model, our framework first identifies and extracts knowledge residing in its parameters. Subsequently, using the extracted knowledge, we construct a downstream task on which we finetune the model. Finally, we measure knowledge utilization based on its performance on the downstream task. By constructing the task based on the model's pre-existing knowledge, we ensure that (1) the model is evaluated solely on its possessed knowledge, avoiding penalties for lacking information and (2) successful task completion relies explicitly on utilizing the model's parametric knowledge, eliminating the insufficient training sig-

nal issue and dataset shortcuts.

In this paper, we provide the first instantiation of this paradigm based on encyclopedic knowledge facts and conduct an extensive study to measure knowledge utilization of PLMs across a wide range of parametric scales (ranging from 125M to 13B). We observe the following:

- PLMs show two different but equally important gaps: (1) The gap in the acquired knowledge and (2) and the gap in parametric knowledge that can be actively applied to downstream tasks (Section 3).

- PLMs are not robust to finetuning distribution shifts, and failure to utilize knowledge worsens with such shifts, further questioning their generalization capabilities (Section 4).

- Although scaling the number of parameters helps to close the first gap, the second still remains in larger sizes (Section 5).

In the next sections, we first describe our framework and its instantiation in detail (Section 2), and finally present our experimental results in Sections 3 to 5.

## 2 Framework

### 2.1 eXtract, TRain, and EVALuate

**Principles** The primary objective of our evaluation framework is to measure how much of the knowledge present in the model's parameters is actually usable in downstream tasks. Ideally, downstream tasks must be designed in a way that solely attributes any success to the model's knowledge being used, while ensuring that failure in performing the task is not due to a lack of pre-training knowledge.

**The Paradigm** To this end, we propose EX-TRACT, TRAIN, AND EVALUATE , which consists of three main steps:

*Step 1.* Given a pre-trained model $\mathcal{M}_\theta$ with parameters $\theta$ and a diagnostic dataset $\mathcal{D}$ (e.g. a set of encyclopedic facts or coding problems), we first extract and identify parametric knowledge as a set of data instances $x \in \mathcal{D}$ the model can solve without further training (zero-shot). We denote such a set as $\mathcal{D}^\theta$, a realization of $\mathcal{M}_\theta$'s parametric knowledge w.r.t $\mathcal{D}$.

*Step 2.* We construct a downstream task $\mathcal{K}$ around the model's own knowledge $\mathcal{D}^\theta$ (e.g. fact

retrieval or following instructions in coding) such that the model can only solve the task by utilizing the knowledge identified in the first step. More formally, we create $\mathcal{K}^\theta_{\text{train}}$ and $\mathcal{K}^\theta_{\text{test}}$ as the non-overlapping train and test sets of downstream task $\mathcal{K}$, where the model learns the task from $\mathcal{K}^\theta_{\text{train}}$.

*Step 3.* Finally, the performance on the test set $\mathcal{K}^\theta_{\text{test}}$ is used as a measure of the model's ability to utilize its knowledge.

Constructing the downstream task based on the model's knowledge ensures that the model is not evaluated on the knowledge it did not acquire during pre-training. Also, the I.I.D. nature of this paradigm (i.e. the model is only exposed to inputs it is already familiar with) allows us to measure whether the model can utilize its knowledge *at all*.

### 2.2 Encyclopedic Knowledge

Factual parametric knowledge as in encyclopedic facts is well-studied in PLMs (Petroni et al., 2019; Jiang et al., 2020) and allows for an objective and systematic evaluation of our framework (Figure 2). Therefore, in this paper, we instantiate XTRAEVAL to measure the utilization of parametric knowledge concerning encyclopedic facts. In this case, the diagnostic dataset $\mathcal{D}$ is a set of encyclopedic facts $\mathcal{D} = \{\langle \mathbf{h}, \mathbf{r}, \mathbf{t} \rangle_i\}_{i=1}^n$ acquired from an off-the-shelf knowledge base (e.g. Wikipedia). Each fact $x_i \in \mathcal{D}$ is a tuple of the form $\langle \text{head}, \text{relation}, \text{tail} \rangle$, such as $\langle \text{Barack Obama}, \text{GraduatedFrom}, \text{Harvard} \rangle$.

In the extraction phase, a pre-trained model $\mathcal{M}_\theta$ has to zero-shot predict the tail entity $t$ given the head entity $h$ and the relation $r$. We use soft-prompting (Qin and Eisner, 2021) to obtain the model's predictions, as it enhances prediction consistency compared to discrete prompts, particularly for moderate-sized models. The extracted knowledge $\mathcal{D}^\theta \subset \mathcal{D}$ is the subset of tuples the model can predict correctly.

Our downstream task $\mathcal{K}$ is a standard document retrieval task (Karpukhin et al., 2020). Given a query $q$, the model retrieves the relevant document from a set of candidates. We construct $\mathcal{K}^\theta$ from the extracted knowledge in $\mathcal{D}^\theta$ by converting each fact $x \in \mathcal{D}^\theta$ into a retrieval instance $k \in \mathcal{K}^\theta$. This conditions the downstream task on the model's knowledge. The conversion generates a query $q$ by removing the tail entity $\mathbf{t}$ from $x$. It then generates relevant and irrelevant documents using a stochastic generator

$$d \sim \mathrm{P}(d \mid H = h, R = r, T = t), \quad (1)$$

| **Encyclopedic Fact:** $x = \langle \mathbf{h}, \mathbf{r}, \mathbf{t} \rangle = \langle \text{Barack Obama}, \text{GraduatedFrom}, \text{Harvard} \rangle$ | |
|---|---|
| **Input** | **Sampled Document** |
| $(\mathbf{h}, \mathbf{r}, \mathbf{t})$ | Barack Obama graduated from Harvard. *Gold document ($d^+$)* |
| $(\mathbf{h}, \mathbf{r}, \cdot)$ | Barack Obama earned a degree from Stanford. *Randomly replacing the tail entity.* |
| $(\cdot, \mathbf{r}, \mathbf{t})$ | Bill Gates received his degree from Harvard. *Randomly replacing the head entity.* |
| $(\mathbf{h}, \cdot, \mathbf{t})$ | Barack Obama was born in Harvard. *Randomly replacing the relation.* |
| $(\cdot, \cdot, \mathbf{t})$ | Steve Jobs died in Harvard. *Keeping the tail entity and sampling others entities.* |
| $(\cdot, \mathbf{r}, \cdot)$ | McGill is the alma mater of Justin Trudeau. *Keeping the relation and sampling others entities.* |
| $(\mathbf{h}, \cdot, \cdot)$ | Barack Obama is located in London. *Keeping the head entity and sampling others entities.* |
| $(\cdot, \cdot, \cdot)$ | Michael jordan was a football player by profession. *Unconditional sampling.* |

Table 1: All possible inputs to the document generator $\mathrm{P}(d \mid H, R, T)$ per each fact $x$ and examples of the corresponding sampled documents. The dot means that the corresponding entity or relation is not given, and the document generator will randomly choose it from $\mathcal{D}^\theta$. The gray text provides an explanation of the sampled document. Note that we do not force the document generator to generate a factual document and the model itself has to predict the relevancy of each document.

where $d$ depends on the head entity $h$, relation $r$, and tail entity $t$. The document generator, $\mathrm{P}(d \mid \cdot)$, selects a template at random and fills in the blanks with the input entities. If $H$, $R$, or $T$ are missing, the generator chooses a random entity from $\mathcal{D}^\theta$ to complete the input. Specifically, we generate relevant document $d^+$ by sampling from $\mathrm{P}(d \mid \cdot)$ with gold entities in $x$ as input, and create irrelevant documents $d^-$ by omitting one or more entities. Each $k$ comprises a tuple $(q, \{d^+, d_1^-, \ldots, d_m^-\})$, where $m$ is the number of irrelevant documents.

We partition $\mathcal{D}^\theta$ randomly (60%-40%) to generate $\mathcal{K}_{\text{train}}^\theta$ and $\mathcal{K}_{\text{test}}^\theta$, which serve as the training and test sets for the downstream task, respectively. We finetune the model on $\mathcal{K}_{\text{train}}^\theta$ in cross-encoder setup (Nogueira and Cho, 2020) with the InfoNCE objective (van den Oord et al., 2019):

$$\mathcal{L}(k) = -\log \frac{\exp(\text{sim}(q, d^+))}{\sum_{d \in \{d^+, d_1^-, \ldots, d_m^-\}} \exp(\text{sim}(q, d))}.$$

The similarity score $\text{sim}(.,)$ is computed as

$$\text{sim}(q, d) = h(\mathcal{M}_\theta([\text{CLS}]; q; d)),$$

where $h$ is a randomly initialized value head that takes the representation of the [CLS] token (or the last token for decoder-only models) and outputs a scalar as the similarity measure (Figure A.1). Finally, we evaluate the model on $\mathcal{K}_{\text{test}}^\theta$ by measuring its accuracy in retrieving the relevant document $d^+$ among $\{d^+, d_1^-, \ldots, d_m^-\}$ for a given query $q$.

The task design ensures that the association between knowledge query $q_i$ and gold fact document $d_i^+$ relies solely on the parametric knowledge represented by $x_i \in \mathcal{D}^\theta$ This is because other variables, like text overlap, are randomly sampled from the same distribution for both query and documents.

Thus, the model can only solve the task by utilizing its internal knowledge. Finetuning on $\mathcal{K}_{\text{train}}^\theta$ should only trigger the utilization of the parametric knowledge.

**Training** The document generator $\mathrm{P}(d \mid \cdot)$ can generate various types of documents for each fact $x \in \mathcal{D}^\theta$. Please refer to Table 1 for a list of all the types. For training, we use three types for *negative* documents $d^-$'s with uniform weights: $(\mathbf{h}, \mathbf{r}, \cdot)$, $(\cdot, \mathbf{r}, \mathbf{t})$, and $(\mathbf{h}, \cdot, \mathbf{t})$ as they are the hardest ones since they only differ in one entity from the query. To keep the GPU memory usage under control, we sample four documents per each type (refer to Section 3.1 for the effect of the number of negatives on the results), which results in a total of $m = 12$ negatives. We resample the documents on each epoch to avoid overfitting and use a validation set to choose the best checkpoint. Also, we keep the learning rate low and use no weight decay to prevent any forgetting. We use three seeds for the extraction phase, three seeds for splitting $\mathcal{D}^\theta$ into train and test, and three seeds for finetuning on the downstream task, which results in 27 different runs per each model.

**Inference** During inference, the model must identify the gold document $d^+$ amidst distractor documents $d^-$'s. To ascertain that the model genuinely recognizes the correct answer, we employ a varied assortment of distractors. Initially, we use document type $(\mathbf{h}, \mathbf{r}, \cdot)$, ensuring all non-gold tails are included. Subsequently, we utilize the remaining non-gold document types listed in Table 1 as distractors, sampling 50 documents for each type. Lastly, we also sample 50 irrelevant but factually correct documents from the test set to assess the model's sensitivity to factual accuracy. We evaluate pre-trained models across various families: OPT (Zhang et al., 2022), GPT-Neo (Black et al., 2021), RoBERTa (Liu et al., 2019), and

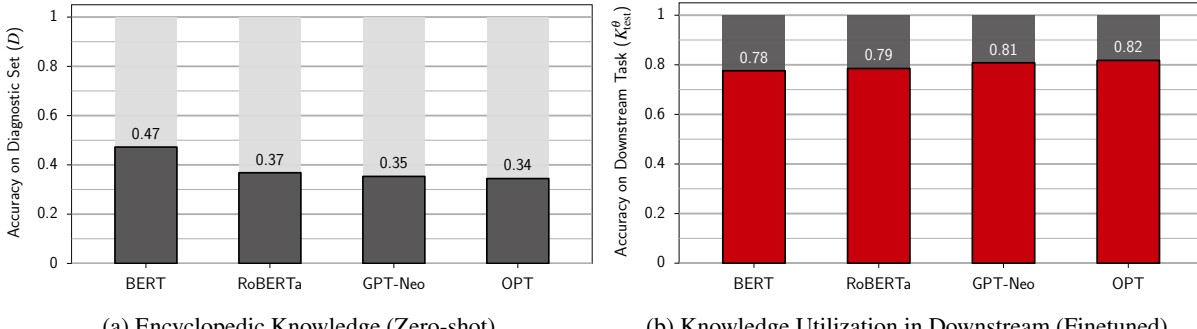

(a) Encyclopedic Knowledge (Zero-shot)  (b) Knowledge Utilization in Downstream (Finetuned)

Figure 3: **(a)** The fraction of encyclopedic facts the pre-trained LM can predict correctly without any training. Reported over three seeds (standard deviation $\sigma \leq 0.004$ for all models). **(b)** The model performance in downstream task (created based on correctly predicted facts) measured as top-1 retrieval accuracy. Averaged over 27 runs per each model ($\sigma \leq 0.011$ for all models). Refer to Appendix B for detailed results.

BERT (Devlin et al., 2019). Unless otherwise stated, we use the base size (125M) of these models. We investigate the scaling behavior of OPT and LLaMa (Touvron et al., 2023) in Section 5. We initialize the diagnostic dataset $\mathcal{D}$ from LAMA (Petroni et al., 2019), which has $34K$ facts over 40 relations. Our results are reported over 1134 finetuning runs (Refer to Appendix A for detailed hyperparameters.)

## 3 Evaluating the Knowledge Utilization

We separately report the fraction of facts ($\mathcal{D}$) that can be extracted and the downstream performance of models in Figure 3.

First, we find that, on par with previous work (Qin and Eisner, 2021), there is a significant gap in the encyclopedic facts the models can correctly predict and the entire facts present in the diagnostic dataset $\mathcal{D}$ (Figure 3a). Note that one can arbitrarily increase the number of correctly predicted by considering a prediction as correct if the gold entity is among the model's top-$k$ predictions. However, we only consider $k = 1$ to only focus on the facts that the model can confidently predict. Nonetheless, we find that BERT and RoBERTa extract slightly more encyclopedic facts than GPT-Neo and OPT.

Critically, all models demonstrate a pronounced gap in downstream task performance, or knowledge utilization, (Figure 3b). This unexpected outcome occurs despite the downstream task being seemingly simple since (1) models are trained and evaluated on examples based on their accurate encyclopedic knowledge predictions, and (2) both $\mathcal{K}_{\text{train}}^{\theta}$ and $\mathcal{K}_{\text{test}}^{\theta}$ are sampled from the same distributions (I.I.D), so the models only encounter seen entities. Notably, OPT and GPT-Neo manage to outperform

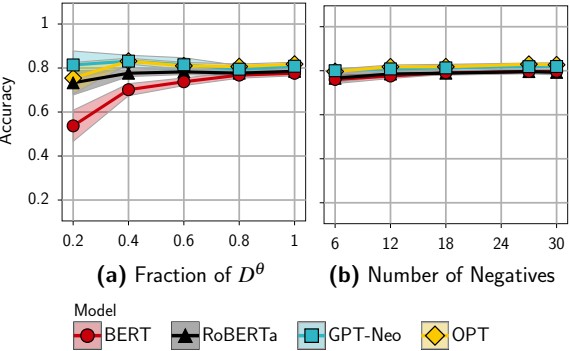

(a) Fraction of $D^{\theta}$  (b) Number of Negatives

Model: BERT  RoBERTa  GPT-Neo  OPT

Figure 4: **(a)** Knowledge utilization when using different fractions of parametric knowledge to create the downstream task. **(b)** The effect of number of negative training documents ($d^-$'s) used for creating the downstream task.

BERT and RoBERTa by a small margin.

This finding suggests that models struggle to utilize their entire parametric knowledge in downstream tasks. In the next sections, we investigate the potential causes of this gap.

### 3.1 Role of Downstream Training Data

**The effect of initial knowledge $\mathcal{D}^{\theta}$** As we utilize $\mathcal{D}^{\theta}$ to create the downstream task, examining the impact of its size ($|\mathcal{D}^{\theta}|$) on knowledge utilization is crucial. If consistent behavior is observed for different sizes, it implies that the utilization gap does not stem from the amount of initial knowledge and must be rooted in inductive biases (e.g., the model or finetuning process), allowing us to measure and compare utilization with different initial knowledge.

To measure such effect, for each model, we first compute $\mathcal{D}^{\theta}$, and then instead of directly using it for $\mathcal{K}^{\theta}$, we sub-sample smaller sets of it at various

fractions and construct the downstream task using each sub-sampled $\mathcal{D}^\theta$. In Figure 4.a, we observe the knowledge utilization is fairly consistent (at least for fractions $> 0.4$) across different sizes of $\mathcal{D}^\theta$ for all models. Larger fractions seem to have less variance as well. This suggests that the utilization performance is intrinsic to the downstream knowledge transfer rather than the initial knowledge residing in the model.

**The effect of the number of negatives** The model learns to apply its parametric knowledge by optimizing the retrieval objective. To ensure the training signal, produced by contrastive loss on $\mathcal{K}^\theta_{\text{train}}$, is strong, we vary the number of negative documents for creating $\mathcal{K}^\theta_{\text{train}}$. If the training signal is weak, we expect knowledge utilization to improve with more negatives.

To answer this question, we follow the same setup as described in Section 2 and increase the number of negative documents sampled per type from 4 to 10. We also consider reducing it to 2 negatives per type to better understand its effectiveness. We keep the initial knowledge $\mathcal{D}^\theta$ fixed.

Figure 4.b summarizes our findings. Knowledge utilization remains the same for all models as we increase the number of negatives. This pattern is observed even with as few as two negatives per type. This suggests that the training signal is strong enough across the board and the gap in knowledge utilization is not rooted in the training objective.

### 3.2 Gap 1 vs. Gap 2

Findings in Section 3.1 shows that the gap in knowledge utilization (i.e. accuracy on $\mathcal{K}^\theta_{\text{test}}$) does not depend on the size of $\mathcal{D}^\theta$ and is fairly consistent across different number of negatives. Moreover, we find that the variation across the random splitting of $\mathcal{D}^\theta$ to create train and test sets of the downstream task is negligible.

The robustness to such design choices allows us to define *Usable Knowledge*, which basically indicates the portion of facts from $\mathcal{D}$ that the model can *actually* utilize in the downstream task. We compute this metric by multiplying the accuracy on $\mathcal{K}^\theta_{\text{test}}$ by the fraction of correctly predicted facts in $\mathcal{D}$. We report the results in Figure 5.

These results clearly demonstrate that there exist two gaps in the models' knowledge. Gap 1 is in how many facts the model knows after pre-training. Gap 2 is in how many of facts the model knows can be truly utilized in downstream tasks. Indeed,

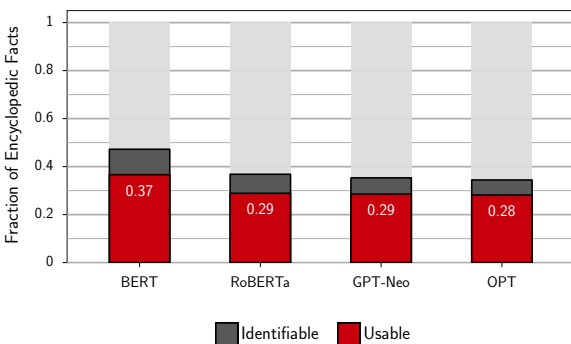

Figure 5: **Gaps in parametric knowledge** ⬜ Gap 1 represents the missing facts in parametric knowledge $\mathcal{D}^\theta$ (what the model knows). ⬛ Gap 2 exists in how many of the known facts the model can actually utilize in downstream tasks (the usable knowledge).

we see that although RoBERTa manages to extract more facts than GPT-Neo, due to Gap 2, it performs the same as GPT-Neo in downstream tasks.

## 4 Robustness of Knowledge Utilization

We intentionally design the downstream task $\mathcal{K}^\theta$ to be straightforward and free of any distributional shift as we want to measure the *maximum* knowledge utilization of the model. However, in real-world applications, it is likely that the model encounter samples that are different from the training distribution. In this section, we investigate the robustness of knowledge application in the presence of such distributional shifts.

### 4.1 Non-I.I.D. $\mathcal{K}^\theta_{\text{train}}$ and $\mathcal{K}^\theta_{\text{test}}$

Recall that we randomly divide $\mathcal{D}^\theta$ into two sets as the data source for the creation of $\mathcal{K}^\theta_{\text{train}}$ and $\mathcal{K}^\theta_{\text{test}}$. In this experiment, however, we split $\mathcal{D}^\theta$ such that the relation types ($\mathbf{r}$) in $\mathcal{K}^\theta_{\text{train}}$ and $\mathcal{K}^\theta_{\text{test}}$ are disjoint. Specifically, we randomly select 60% of the relations and their corresponding facts for $\mathcal{K}^\theta_{\text{train}}$ and the rest for $\mathcal{K}^\theta_{\text{test}}$. We repeat this process over three seeds to create three different splits. We still follow the same procedure for converting knowledge triples to document retrieval examples as explained in Section 2. In this way, we ensure we don't change the task's nature, i.e. the model still needs to apply its parametric knowledge to solve the task, but the distributional shift between $\mathcal{K}^\theta_{\text{train}}$ and $\mathcal{K}^\theta_{\text{test}}$ can represent real-world scenarios. If the model learns to systematically apply its knowledge, we expect its downstream performance to be similar to or close to the I.I.D. setting (Section 3).

We observe downstream task performance drops

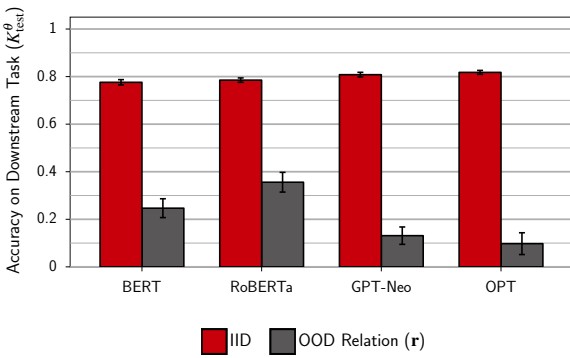

Figure 6: **Robustness to distributional shift** In the OOD setting, we produce a distributional shift (over the relation types) between the examples in the train and test set of the downstream task $\mathcal{K}^\theta$. All models fail to generalize to unseen relations. The IID setting is the same as the one described in Section 2 and repeated from Figure 3b for comparison.

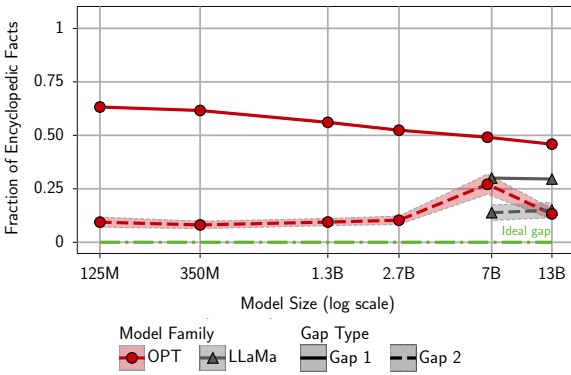

Figure 7: **Gaps in parametric knowledge** Knowledge gaps directly compute across different model sizes. Specifically, we use $1 - (\text{Accuracy on } \mathcal{D}^\theta)$ for Gap 1 and $(\text{Accuracy on } \mathcal{D}^\theta) \times (1 - \text{downstream accuracy})$ for Gap 2[†].

significantly for all models when evaluated OOD (Figure 6). This indicates the models cannot use their knowledge on examples with unseen relation types, though all relations and facts originate in $\mathcal{D}^\theta$. Thus, knowledge usage in downstream tasks is sensitive to distribution shifts, suggesting failure to apply pre-training knowledge may be more severe in real-world applications.

## 5 Effect of Scaling law On The Gaps

Recent NLP success has come from scaling up pre-training model parameters (Brown et al., 2020). With larger models and increased compute, capabilities such as in-context learning and chain-of-thought reasoning emerge (Wei et al., 2022b). The expanded capacity allows these models to absorb more knowledge from pre-training data, improving their usefulness as knowledge sources. However, it remains uncertain if scaling boosts the proportion of pre-training knowledge applicable to downstream tasks. Ideally, we like to see a narrowing gap in pre-training knowledge alongside superior knowledge utilization.

To investigate this, we evaluate XTRAEVAL on increasing sizes of OPT and LLaMa models. Specifically, at each scale, we first extract the model's parametric knowledge and then create the downstream task based on it using the same procedure as described in Section 2. Figure 1 reports the results of this experiment.

First, we confirm that a greater fraction of knowledge triples in $\mathcal{D}$ can be identified in larger models,

suggesting they acquire more knowledge from pre-training data. Secondly, we find that the gap between identifiable and usable knowledge persists in larger models, and their ability to apply knowledge in downstream tasks does not improve with scaling. Figure 7 illustrates these gaps directly, demonstrating that while Gap 1 decreases in larger models, Gap 2 remains relatively unchanged.

The results suggest that while PLMs, even at small scales, pose considerable knowledge, extracting an equivalent amount of usable knowledge necessitates much larger models. For instance, OPT-125M accurately predicts 34% of encyclopedic facts, but only OPT-13B (approximately $100\times$ larger) can reliably apply the same volume in downstream tasks. Enhanced pre-training routines, including the use of more data or higher quality data, can bolster knowledge acquisition, as is clearly demonstrated by LLaMa models. Notably, LLaMa-7B significantly outperforms OPT-13B. While LLaMa models possess a greater amount of usable knowledge due to superior initial knowledge, a gap in knowledge utilization remains discernible (Figure 7).

## 6 Discussion

Lately, pre-trained language models with chatbot interfaces have increasingly been served as knowledge bases (Ouyang et al., 2022). These chatbots typically employ the model's parametric

---

[†]We conducted the experiments for OPT-6.7B multiple times, and observed a dip in performance in all runs. We suspect that this consistent decline may be attributed to issues that arose during the pre-training phase.

knowledge to respond to queries and offer information. Our study examines the dependability of this knowledge and its impact on downstream task performance. We discover that, regardless of inductive biases, PLMs face difficulty utilizing their full knowledge in downstream tasks (Section 3). This unreliability of parametric knowledge could constrain the concept of "PLMs as differentiable knowledge bases."

Additionally, our findings show that the utilization gap persists even with scaling (Section 5). Notably, while models at each scale capture more knowledge from pre-training data, obtaining the same amount of usable knowledge requires significantly larger models. This exposes a potential constraint in the recent trend of adopting mid-sized PLMs (Li et al., 2023).

Lastly, we discover that knowledge utilization depends on the peculiarities of finetuning data for downstream tasks. Specifically, as seen in Section 4, PLMs struggle to apply their knowledge to relation types not encountered during finetuning, even if they accurately predicted such facts in step 1. This generalization gap could highlight challenges within the recent SFT-RLHF paradigm (Ouyang et al., 2022). For instance, the model may only adhere to instructions and excel at tasks resembling the finetuning data. Consequently, it might be necessary to meticulously craft finetuning data to activate and utilize all aspects of parametric knowledge in downstream tasks. However, it requires elaborate studies to establish the systematic issues in knowledge application beyond encyclopedic knowledge like procedural and task knowledge.

## 7    Related Work

**Parametric Knowledge**    Petroni et al. (2019) constructed a probing dataset to measure the factual knowledge present in PLMs. They showed that many encyclopedic facts can be extracted without further training of the model and proposed PLMs as a new type of knowledge base, which can be trained on the unstructured text and queried using natural language. Follow-up work improves the methods for probing and extracting world knowledge from PLMs (Jiang et al., 2020; Shin et al., 2020; Qin and Eisner, 2021; Newman et al., 2022). Apart from encyclopedic facts, studies have explored PLMs' parametric knowledge in other areas, such as linguistic structures (Tenney et al., 2019b; Blevins et al., 2022), and commonsense (Zhou et al., 2020;

Liu et al., 2022a). Recently, the emergent abilities of LLMs have shown that they acquire skills like coding (Chen et al., 2021), reasoning (Chowdhery et al., 2022), and in-context learning (Brown et al., 2020), in addition to the previously mentioned knowledge.

**Using the Parametric Knowledge**    Roberts et al. (2020) finetune a pre-trained T5 model for question answering in a closed-book setting and showed that it can perform on par or better than models that use explicit knowledge bases. Wang et al. (2021) made a similar observation for the BART model. More recently, PLMs are being used to generate facts and documents for knowledge-intensive tasks (Li et al., 2022; Liu et al., 2022b; Yu et al., 2023). In this paradigm, in order to answer factual questions, instead of retrieving relevant documents, the model has to first generate the facts and then answer the question with those facts as context. This paradigm shows that the models may not be able to use their parametric knowledge on their own and need explicit grounding to be able to use it. Furthermore, there is a plethora of work that investigates whether the model employs its linguistic knowledge when solving downstream language understanding tasks. McCoy et al. (2019) shows that RoBERTa does not use its linguistic knowledge for solving NLI. Instead, it relies on shallow heuristics. Lovering et al. (2021)'s observation aligns with this finding and shows the training data used for the downstream task needs to have enough evidence to trigger the model's linguistic knowledge. In our work, we use a more general notation of parametric knowledge and investigate utilization in cases where sufficient evidence is present in the finetuning data.

## 8    Conclusion

In this study, we presented EXTRACT, TRAIN, AND EVALUATE (XTRAEVAL ), a framework designed to assess the parametric knowledge of pre-trained language models. Employing XTRAEVAL we identified a previously unnoticed gap in what models know and how much of it they can actually use. Our findings reveal that this gap exists not only in smaller models but also persists in larger ones. Additionally, we demonstrate that a distributional shift in finetuning data can result in even larger gaps between the model's knowledge and its practical application in downstream tasks.

## Limitations

Although XTRAEVAL is agnostic to the specific type of parametric knowledge, our work primarily focuses on encyclopedic facts as one type of world knowledge that PLMs can acquire. It is plausible that similar results would hold for other knowledge types, however, further work is needed for a precise investigation.

While there are various downstream tasks that could be evaluated, we primarily focus on document retrieval as it allows us to systematically demonstrate the key issue of knowledge application that we aim to highlight. We also acknowledge that our study was limited to a few model families and parameter scales due to compute constraints. However, our evaluation protocol is model agnostic, enabling future work to explore this phenomenon on other tasks and with different models.

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

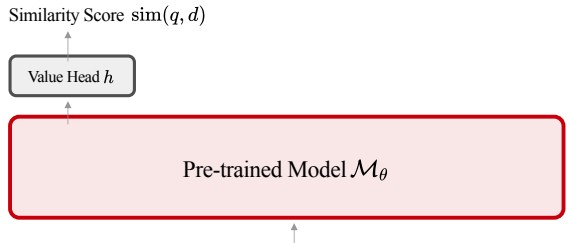

Similarity Score $\text{sim}(q, d)$

"[CLS] Where did Obama graduate from? Obama graduated from Harvad."

Figure A.1: Cross-encoder document retrieval setup (Nogueira and Cho, 2020). For decoder-only models, the value head takes the representation of the last input token.

## A    Training Details

### A.1    Knowledge Extraction

We adopt the same procedure as outlined by Qin and Eisner (2021) for extracting knowledge facts from a frozen PLM. Specifically, we utilize soft-prompts instead of discrete prompts. We insert three soft prompts before and after the head entity and allocate distinct soft-prompts for each relation type. We then train them using the training set provided by Qin and Eisner (2021) and employ a validation set to select the best checkpoint. The hyperparameters used in this stage, borrowed from Qin and Eisner (2021), are summarized in Table A.1.

### A.2    Finetuning

For fine-tuning the models, we follow a straightforward procedure, training the models in a cross-encoder setup (Figure A.1). The hyperparameters used for fine-tuning are listed in Table A.2. In initial experiments, we tried $\text{lr} \in 1 \times 10^{-5}, 3 \times 10^{-5}, 5 \times 10^{-5}$, but we did not find any significant difference between them for all models. Thus, we opted to use the same learning rate for all models.

### A.3    Dataset Details

We employ the LAMA dataset (Petroni et al., 2019) as our diagnostic set, consisting of 34,039 facts. The training and validation sets for soft-prompt training, provided by Qin and Eisner (2021), contain 29,029 and 7,255 triples, respectively. The size of the downstream dataset varies from one model instance to another, as we construct the downstream task based on its extracted knowledge. The size of such datasets can be easily calculated by multiplying the number of facts in $\mathcal{D}$ by the knowledge

| Parameter | Value |
|---|---|
| Optimizer | AdamW |
| Learning rate | $1 \times 10^{-4}$ |
| Weight Decay | 0 |
| Batch size | 64 |
| Learning Rate Scheduler | Polynomial |
| Warm Up | 6% of training steps |
| # Train Epochs | 20 |

Table A.1: Summary of hyperparameters used in knowledge extraction stage (stage 1).

| Parameter | Value |
|---|---|
| Optimizer | AdamW |
| Learning rate | $1 \times 10^{-5}$ |
| Weight Decay | 0 |
| Batch size | 32 |
| Learning Rate Scheduler | Polynomial |
| Warm Up | 6% of training steps |
| # Train Epochs | 20 |

Table A.2: Summary of hyperparameters used in fine-tuning on downstream task (stage 2).

extraction accuracy obtained by the model.

### A.4    Scaling Experiment Details

We adhere to the same procedure as other models in the scaling experiments of OPT and LLaMa. For larger models, we only increase the batch size following Iyer et al. (2023) to ensure better fine-tuning stability. Due to the extensive computational and cost requirements of fully fine-tuning the 7B and 13B models, we limit the number of seeds for these variants. Specifically, we trained four seeds for LLaMa-7B, two for OPT-13B, and one for LLaMa-13B.

### A.5    Computational Resources

For the experiments in this study, we exclusively use NVIDIA V100-32GB GPUs. Models with $\leq$ 350M parameters were trained on a single GPU. For knowledge extraction, we utilized parallel training implemented by DeepSpeed for larger parameter sizes. Table A.4 displays the number of GPUs used and the approximate duration of a single training run.

### A.6    Reproducibility

For the experiments in this study, we exclusively use NVIDIA V100-32GB GPUs. Models with $\leq$ 350M parameters were trained on a single GPU. For knowledge extraction, we utilized parallel train-

| Model | Knowledge Extraction | Downstream Finetuning |
|---|---|---|
| | OPT | |
| 125M | 64 | 32 |
| 350M | 64 | 32 |
| 1.3B | 64 | 32 |
| 2.7B | 64 | 32 |
| 6.7B | 128 | 64 |
| 13B | 128 | 64 |
| | LLaMa | |
| 7B | 128 | 64 |
| 13B | 128 | 64 |

Table A.3: Batch size used in the scaling experiments.

| Model | Knowledge Extraction | Downstream Finetuning |
|---|---|---|
| | OPT | |
| 125M | 1 (0h 30m) | 1 (0h 30m) |
| 350M | 1 (0h 30m) | 1 (1h 0m) |
| 1.3B | 2 (1h 40m) | 2 (1h 40m) |
| 2.7B | 4 (3h 30m) | 4 (3h 50m) |
| 6.7B | 8 (1h 20m) | 8 (8h 40m) |
| 13B | 8 (5h 20m) | 8 (1d 4h 40m) |
| | LLaMa | |
| 7B | 8 (8h 30m) | 8 (17h 40m) |
| 13B | 8 (17h 30m) | 8 (1d 15h 15m) |

Table A.4: Number of GPU and approximate training time for each model size.

ing implemented by DeepSpeed for larger parameter sizes. Table A.4 displays the number of GPUs used and the approximate duration of a single training run.

## B  Full Results

### B.1  Detailed IID Results

We present the comprehensive results of our experiments from Section 3 in Tables B.5 to B.8.

### B.2  Detailed OOD Results

Table B.9 provides a detailed account of the experiment results in Section 4.

### B.3  Detailed Scaling Results

The detailed results of Section 5 are presented in Tables B.10 and B.11.

| Model | Knowledge Extraction Accuracy | Downstream Accuracy |
|---|---|---|
| BERT | $0.4722 \pm 0.0006$ | $0.7760 \pm 0.0112$ |
| RoBERTa | $0.3681 \pm 0.0007$ | $0.7852 \pm 0.0092$ |
| GPT-Neo | $0.3531 \pm 0.0005$ | $0.8081 \pm 0.0099$ |
| OPT | $0.3444 \pm 0.0038$ | $0.8177 \pm 0.0083$ |

Table B.5: Mean±standard deviation of results presented in Figure 3a and Figure 3b Each number is computed over 27 runs and models are in 125M parameter regime.

| Model | Downstream Accuracy (Per Number of Negatives) | | | | |
|---|---|---|---|---|---|
| | 6 | 12 | 18 | 27 | 30 |
| BERT | $0.7604 \pm 0.0190$ | $0.7760 \pm 0.0112$ | $0.7908 \pm 0.0075$ | $0.7965 \pm 0.0081$ | $0.7970 \pm 0.0055$ |
| RoBERTa | $0.7660 \pm 0.0192$ | $0.7852 \pm 0.0092$ | $0.7889 \pm 0.0086$ | $0.7961 \pm 0.0100$ | $0.7927 \pm 0.0117$ |
| GPT-Neo | $0.8000 \pm 0.0115$ | $0.8081 \pm 0.0099$ | $0.8129 \pm 0.0128$ | $0.7577 \pm 0.2186$ | $0.7586 \pm 0.2189$ |
| OPT | $0.7966 \pm 0.0109$ | $0.8177 \pm 0.0083$ | $0.8179 \pm 0.0113$ | $0.8294 \pm 0.0054$ | $0.8269 \pm 0.0095$ |

Table B.6: Mean±standard deviation of results presented in Figure 4. Each number is computed over 27 runs and models are in 125M parameter regime.

| Model | Downstream Accuracy (Per Fraction of $\mathcal{D}^\theta$) | | | | |
|---|---|---|---|---|---|
| | 0.2 | 0.4 | 0.6 | 0.8 | 1 |
| BERT | $0.5371 \pm 0.0707$ | $0.7015 \pm 0.0272$ | $0.7376 \pm 0.0173$ | $0.7678 \pm 0.0138$ | $0.7760 \pm 0.0112$ |
| RoBERTa | $0.7329 \pm 0.0554$ | $0.7763 \pm 0.0153$ | $0.7824 \pm 0.0122$ | $0.7769 \pm 0.0145$ | $0.7852 \pm 0.0092$ |
| GPT-Neo | $0.8139 \pm 0.0633$ | $0.8313 \pm 0.0268$ | $0.8149 \pm 0.0318$ | $0.7948 \pm 0.0164$ | $0.8081 \pm 0.0099$ |
| OPT | $0.7542 \pm 0.0710$ | $0.8312 \pm 0.0135$ | $0.8100 \pm 0.0158$ | $0.8065 \pm 0.0119$ | $0.8177 \pm 0.0083$ |

Table B.7: Mean±standard deviation of results presented in Figure 4. Each number is computed over 27 runs and models are in 125M parameter regime.

| Model | Fraction of Encyclopedic Facts | | Knowledge Gaps | |
|---|---|---|---|---|
| | Identifiable | Usable | Gap 1 | Gap 2 |
| BERT | $0.4722 \pm 0.0006$ | $0.3665 \pm 0.0052$ | $0.5277 \pm 0.0006$ | $0.1058 \pm 0.0053$ |
| RoBERTa | $0.3681 \pm 0.0007$ | $0.2890 \pm 0.0035$ | $0.6319 \pm 0.0008$ | $0.0791 \pm 0.0034$ |
| GPT-Neo | $0.3531 \pm 0.0005$ | $0.2854 \pm 0.0035$ | $0.6468 \pm 0.0006$ | $0.0678 \pm 0.0035$ |
| OPT | $0.3444 \pm 0.0038$ | $0.2816 \pm 0.0036$ | $0.6556 \pm 0.0038$ | $0.0628 \pm 0.0030$ |

Table B.8: Full results including mean±standard deviation of experiments presented in Figure 5.

| | Downstream Accuracy | | | |
|---|---|---|---|---|
| | BERT | RoBERTa | GPT-Neo | OPT |
| IID | $0.7760 \pm 0.0112$ | $0.7852 \pm 0.0092$ | $0.8081 \pm 0.0099$ | $0.8177 \pm 0.0083$ |
| OOD Relation (**r**) - All | $0.2467 \pm 0.0396$ | $0.3559 \pm 0.0415$ | $0.1312 \pm 0.0365$ | $0.0976 \pm 0.0458$ |
| OOD Relation (**r**) - Seen entities | $0.2451 \pm 0.0337$ | $0.3778 \pm 0.0474$ | $0.1330 \pm 0.0346$ | $0.1001 \pm 0.0489$ |
| OOD Relation (**r**) - Unseen entities | $0.2361 \pm 0.0608$ | $0.3176 \pm 0.0616$ | $0.1317 \pm 0.0503$ | $0.0971 \pm 0.0499$ |

Table B.9: Full results including mean±standard deviation of experiments presented in Figure 6. Each number is computed over 27 runs.

| Model | Knowledge Extraction Accuracy | Downstream Accuracy |
|---|---|---|
| | OPT | |
| 125M | $0.3677 \pm 0.0045$ | $0.6836 \pm 0.0049$ |
| 350M | $0.3839 \pm 0.0017$ | $0.7451 \pm 0.0053$ |
| 1.3B | $0.4403 \pm 0.0052$ | $0.7485 \pm 0.0026$ |
| 2.7B | $0.4761 \pm 0.0019$ | $0.7458 \pm 0.0149$ |
| 6.7B | $0.5090 \pm 0.0016$ | $0.3819 \pm 0.0220$ |
| 13B | $0.5407 \pm 0.0008$ | $0.7155 \pm 0.0013$ |
| | LLaMa | |
| 7B | $0.6952 \pm 0.0078$ | $0.7548 \pm 0.0099$ |
| 13B | $0.7014 \pm 0.0025$ | $0.7508$ |

Table B.10: Results of stage 1 and stage 2 for various parameter sizes, which is used to present plots in Figures 1 and 7

| | Fraction of Encyclopedic Facts | | Knowledge Gaps | |
|---|---|---|---|---|
| Model | Identifiable | Usable | Gap 1 | Gap 2 |
| | | OPT | | |
| 125M | $0.3677 \pm 0.0045$ | $0.2513 \pm 0.0013$ | $0.6323 \pm 0.0033$ | $0.0941 \pm 0.0234$ |
| 350M | $0.3839 \pm 0.0017$ | $0.2860 \pm 0.0019$ | $0.6161 \pm 0.0015$ | $0.0811 \pm 0.0174$ |
| 1.3B | $0.4403 \pm 0.0052$ | $0.3289 \pm 0.0037$ | $0.5606 \pm 0.0037$ | $0.0945 \pm 0.0166$ |
| 2.7B | $0.4761 \pm 0.0019$ | $0.3551 \pm 0.0077$ | $0.5239 \pm 0.0016$ | $0.1029 \pm 0.0196$ |
| 6.7B | $0.5090 \pm 0.0016$ | $0.1944 \pm 0.0112$ | $0.4910 \pm 0.0014$ | $0.2712 \pm 0.0467$ |
| 13B | $0.5407 \pm 0.0008$ | $0.3874 \pm 0.0007$ | $0.4586 \pm 0.0000$ | $0.1325 \pm 0.0249$ |
| | | LLaMa | | |
| 7B | $0.6952 \pm 0.0078$ | $0.5282 \pm 0.0062$ | $0.3002 \pm 0.0010$ | $0.1382 \pm 0.0358$ |
| 13B | $0.7014 \pm 0.0025$ | $0.5288$ | $0.2957 \pm 0.0000$ | $0.1503 \pm 0.0356$ |

Table B.11: Mean±standard deviation of results presented in Figures 1 and 7.