# OpenReview forum: "Measuring the Knowledge Acquisition-Utilization Gap in Pretrained Language Models"
_EMNLP/2023/Conference — EMNLP 2023 Findings_

### Official Review · Reviewer_GyXC · 2023-08-04

**Soundness:** 4

**Excitement:**

4: Strong: This paper deepens the understanding of some phenomenon or lowers the barriers to an existing research direction.

**Paper Topic And Main Contributions:**

This paper investigates the usage of parametric knowledge from pretrained large language (LLM) models in downstream tasks. It proposes an approach to quantify and extract knowledge from the pretrained LLM, using KB/dataset facts and implement a downstream task based on the extract knowledge. The setup is careful design to assess the impact of only the parametric knowledge on downstream tasks. This is an useful investigation since recently supervised finetuning / instruction tuning / RLHF are used to enable the model to follow instructions based on its parametric knowledge.

Specifically, given a pretrained LLM, the proposed approach:
Extract knowledge present in the model parameters
Construct a downstream task based on the extracted knowledge
Measure knowledge "retrieval" based on the performance in the downstream task.

Experiments indicate that there is a gap in knowledge utilization (downstream performance) suggesting that it can be hard for the models to make use of their parametric knowledge during downstream fine tuned tasks.

The paper conducts an experiment where the model needs to use its parametric knowledge to solve the downstream task with a distributional shift where the model is tested with relations which are not used during fine tuning. It is observed that the performance drops when evaluating in such out-of-domain scenarios and the model cannot employ its knowledge on examples with unseen relations even though they were present in its parametric knowledge before fine tuning.

Finally, the paper investigates whether the scaling (# of model parameters) improves the amount of pretrained knowledge employed during the downstream task and shows that the gap between identifiable and used knowledge remains for larger models.


**Questions For The Authors:**

- Are the prompts generated by the document generator diverse enough to avoid overfitting? The data might lack diversity and hurt the model generalization capabilities (for example, it seems that many examples start with entities). Could the authors provide some thoughts on the relation between the data format used in the finetuning phase and the catastrophic forgetting of pretrained knowledge?
- Is there any specific motivation for why only the relations were used for creating the distributional shift? Could the same be done with the entities to investigate whether the model can maintain the knowledge facts for specific entities?


**Reasons To Accept:**

- This paper proposes a framework to analyse acquisition and utilization of knowledge during pretraining/fine tuning phases in pretraining language models, which might help to shed light on the extent the pretrained knowledge can be leveraged during subsequent finetuning phases.
- The systematic design enables the evaluation of the model solely based on its current learned knowledge, avoiding cases where the model does not have knowledge about the information.


**Reasons To Reject:**

- The paper only evaluates encyclopedic knowledge using one dataset based on wikipedia which might be restrictive to derive conclusive findings. It would be interesting to investigate other datasets (e.g., WebNLG [1]) and types of knowledge (e.g., commonsense [2])
- It seems not not very clear that the robustness experiment (Section 4) would be the accurate way to capture that the model can't use its pretrained knowledge since the fine-tuning procedure might be making the model to overfit learning to catastrophic forgetting. A further investigation could be done with more diverse data and prompts (similar to what instruction tuning does) in order to avoid overfit and to preserve the pretrained knowledge (see questions for the authors).

[1] The WebNLG Challenge: Generating Text from RDF Data (Gardent et al., INLG 2017).
[2] UNICORN on RAINBOW: A Universal Commonsense Reasoning Model on a New Multitask Benchmark  (Lourie et al. 2021).


**Reproducibility:**

4: Could mostly reproduce the results, but there may be some variation because of sample variance or minor variations in their interpretation of the protocol or method.

**Reviewer Confidence:**

4: Quite sure. I tried to check the important points carefully. It's unlikely, though conceivable, that I missed something that should affect my ratings.

---

> ### Author Rebuttal · Authors · 2023-08-29
>
> Dear reviewer,
>
> We are grateful for your meticulous review and insightful feedback on our work. It is especially encouraging to hear your positive remarks, which highlight the **"careful and systematic design of our framework."** We are also pleased that you find the **usefulness** of our investigation compelling, particularly in its connection to the emerging paradigm of **"instruction tuning,"** and its potential to illuminate “the extent the pretrained knowledge can be leveraged.”
>
> We're also encouraged by the supportive feedback from other reviewers. Specifically, Reviewer 1 (STEM) highlighted the **novelty** of our framework for measuring knowledge utilization. Reviewer 2 (twpn) appreciated our “**extensive and comprehensive** experiments and analyses”, a view also shared by Reviewer 1 (STEM) who also found our analysis to be of particular interest. Both reviewers further acknowledged the **impact of our work** in "highlighting an interesting research direction," providing a way to "track progress across different approaches," and offering a "better understanding of knowledge utilization."
>
> We now address the key questions and concerns in detail below:
>
> > Use of Other datasets
>
> We are glad our thought processes match. At the preliminary stage of our research, we did indeed consider using commonsense datasets (e.g. ConceptNet[1]) in the framework (for populating $\mathcal{D}$). However, with further investigation, we encountered challenges with the ambiguity inherent in this type of knowledge. Specifically, entities and relations are often not uniquely mapped.  For example, if the gold triple is (fish, AtLocation, aquarium), another equally correct triple could be (fish, AtLocation, sea). When we use such data, we often find that the model's predictions are actually correct, but do not match the dataset’s answer:
> Query: “Where is fish found?” Gold: “Something you find at the aquarium is a fish”, Model’s Retrieval Result: “Something you find under the Sea is fish.”
>
> Other examples like this include:
> - Query: “What do people have?” Gold: “The people have eyes”, Model’s Retrieval Result: “People have babies.”
> - Query: “What are the vultures?” Gold: “Vultures are scavengers”, Model’s Retrieval Result: “A vulture is a animal”
>
> In light of these observations, we made the deliberate choice to forgo this type of knowledge to avoid introducing a potential confounder into our study.
>
> For encyclopedic facts, we also considered other knowledge bases than DBPedia​​ (Wikipedia’s relational knowledge base), including Freebase, an older but separately crafted knowledge base. But, we found similar patterns (to those of DBPedia). Based on these initial observations, we opted to use a well-established DBPedia-sourced dataset in our study, rather than introducing a new one, and instead we made sure to perform a thorough study with various model architectures, families, and sizes, with many seeds. Nevertheless, we recognize the importance of exploring the findings of this framework in other domains and fully endorse introduction of new dataset and knowledge types as an important avenue for future research.
>
> > Robustness Experiments in Section 4 & Question 2
>
> We appreciate the reviewer’s thoughtful feedback on the experiments in Sec. 4 regarding the potential risk of forgetting. To address this, we like to kindly note that our experiments in Section 4 were designed to *closely* mimic the setup of the experiments in Section 3 (of course except for the distribution of relations). This was done deliberately to make everything the same across both setups (please refer to lines 397-411 for more details), including the same number of negatives, identical hyperparameters, same document generator and so on; the number of examples for training and evaluation also remained approximately equivalent. Therefore, we believe that given these considerations, if significant forgetting were occurring during fine-tuning, we wouldn't expect to see the marked performance gap between the OOD and IID scenarios that our results show. While these methodological choices were intended to carefully isolate the effect of distributional shift, we acknowledge that the fine-tuning process could indeed introduce an element of forgetting. But, we wish to note that our fine-tuning approach adheres to standard practices commonly used in the field. This implies that even if some level of forgetting is present, it actually helps to reinforce the core message of our work, which is to highlight noteworthy issues in methodologies that are commonly accepted within our field.
>
> Regarding your second question, the reviewer makes an astute observation in exploring the distributional shift with entities in addition to relations. Although the idea of segmenting $\mathcal{D}^\theta$ based on specific entities, like the "head" of a triple, is noteworthy, this approach comes with its own set of challenges, primarily due to the sparse availability of knowledge triples for each entity. For instance, the number of triples involving "Barack Obama" as the head entity would be substantially fewer compared to triples with the relation "born_in." This scarcity of data for individual entities makes it difficult to draw robust conclusions about whether the model retains or forgets its pre-trained knowledge concerning specific entities. Thus, we opted to employ the relation for this particular experiment to have reliable evaluation.
>
> > Diverse Document Generation (Question 1)
>
> Thank you for raising the insightful question regarding the diversity of generated prompts and its impact on the forgetting of pretrained knowledge. In our study, we utilized the prompts from the work of Qin et al. [2] and augmented them as necessary. We indeed acknowledge that employing a more diverse set of prompts could be beneficial as such diversity would offer a more faithful representation of real-world scenarios in which users interact with the model. However, we like to highlight that the design of our framework can inherently mitigate the issues that may stem from a lack of prompt diversity. We employ the same document generator for both the training and testing phases, thereby ensuring that the model is exposed to and evaluated on the same set of prompts. This design choice aims to provide an unbiased evaluation environment. Essentially, the model is never tested on prompts it hasn't seen during training. So, even if the model overfits to specific sentence structures, it won't be tested to utilize its knowledge in unfamiliar sentences. This setup allows us to focus on the core question of how well the model applies its pretrained knowledge to the task at hand.
>
>
> - [1] Speer et al, https://arxiv.org/abs/1612.03975, AAAI 2017
> - [2] Qin et al, https://aclanthology.org/2021.naacl-main.410/, NAACL 2021

---

### Official Review · Reviewer_twpn · 2023-08-04

**Soundness:** 3

**Excitement:**

3: Ambivalent: It has merits (e.g., it reports state-of-the-art results, the idea is nice), but there are key weaknesses (e.g., it describes incremental work), and it can significantly benefit from another round of revision. However, I won't object to accepting it if my co-reviewers champion it.

**Paper Topic And Main Contributions:**

This paper introduces XTRAEVAL (EXTRACT, TRAIN, AND EVALUATE) to measure how much parametric knowledge is used in downstream tasks. It provides valuable insights by comparing multiple pre-trained models, enhancing our understanding of knowledge utilization in natural language processing.


**Questions For The Authors:**

1. Have you considered using more comprehensive probe methods in combination?
2. Why did you choose to predict relationships as the probe method for knowledge?
3. Why did you choose document retrieval as the automatically generated downstream task?

**Reasons To Accept:**

1. This paper introduces XTRAEVAL (EXTRACT, TRAIN, AND EVALUATE) to measure how much parametric knowledge is used in downstream tasks.
2. The paper presents extensive and comprehensive experiments and analyses, yielding valuable insights through comparisons of multiple pre-trained models. These insights contribute to a better understanding of knowledge utilization in natural language processing.

**Reasons To Reject:**

The conclusions drawn in the paper would benefit from additional supporting evidence. While detecting knowledge presence in PLMs through relation prediction tasks might be more favorable for BERT-like models, it is essential to explore multiple methods. By employing various approaches and tasks, the study can provide a stronger foundation for the conclusions, ensuring robustness and generalizability of the findings regarding knowledge utilization in pre-trained language models.


**Reproducibility:**

4: Could mostly reproduce the results, but there may be some variation because of sample variance or minor variations in their interpretation of the protocol or method.

**Reviewer Confidence:**

3: Pretty sure, but there's a chance I missed something. Although I have a good feel for this area in general, I did not carefully check the paper's details, e.g., the math, experimental design, or novelty.

---

> ### Author Rebuttal · Authors · 2023-08-29
>
> Dear reviewer,
>
> Thank you for investing time in reviewing our paper and offering valuable feedback. We are encouraged to find that you commend our work for its **"extensive and comprehensive experiments and analyses,"** and acknowledge its contribution towards a **"better understanding of knowledge utilization."**
>
> We are heartened by the unanimous recognition from several reviewers regarding the quality and significance of our work. Reviewer 1 (STEM) kindly acknowledged our framework's **"novelty in testing knowledge utilization"** and found our finding and analysis "interesting". We are grateful to reviewer 3 (GyXC) acknowledging "systematic" and "careful" design of our framework, further increasing our confidence in the **rigor** of our work.Regarding the **impact**, Reviewer 1 (STEM) praised our work for outlining an "interesting research direction" and for **"tracking progress in finetuning methods."** These sentiments were also echoed by Reviewer 3 (GyXC), who highlighted the work's significance in illuminating "the extent to which pretrained knowledge can be leveraged."
>
> We now address the key questions and concerns in detail below:
>
> > Exploration of Other Tasks and Knowledge Types & Question 2
>
> Thank you for your insightful comments on the need for additional exploration of other tasks. We agree with your sentiment that exploring other types of knowledge or tasks would strengthen our work. However, it's worth mentioning that our choices in designing this XTraEVAL were deliberate for several reasons, which lead to the creation of robust yet novel paradigm as highlighted by reviewers 1 (STEM) and 3 (GyXC).
>
> First, one of our primary goals was to create meticulous evaluation protocol that exclusively measures *knowledge utilization* within pretrained language models (PLMs). A necessary definition in such a protocol is a clear and tangible formulation of knowledge itself. As also mentioned in lines 183-186, encyclopedic facts serve a basic, yet fundamental formulation. More importantly, they are a well-established convention in the field and has rich literature around it [1,2,3,4].
>
> Furthermore, the ground truth for these facts is sourced from Wikipedia, a reliable and well-curated repository. This minimizes ambiguity in these knowledge facts (e.g. both `Obama was born in Hawaii` and `Obama was born in <some-other-place>` can’t be true at the same time), thereby allowing us to produce a robust evaluation.
>
> Thirdly, we’d like to note that in our paradigm, encyclopedic facts consider all entities—head and tail—together, forming a cohesive piece of knowledge. While structured as triples (head, relation, tail), these facts are presented to the model in a natural language format, which is quite generic and versatile. This is particularly advantageous for zero-shot knowledge probing, aligning closely with the objectives these PLMs were trained on. Such an approach significantly minimizes any distributional shift between the pretraining and knowledge extraction phases. In leveraging this property, we were indeed able to evaluate a range of models, spanning from encoder-only to decoder-only (autoregressive) architectures, resulting in the comprehensive analysis you kindly acknowledged.
>
> While our investigation is focused, we have taken care to conduct a thorough study. We believe our work could serve as a foundational effort in establishing a framework for measuring knowledge utilization and sets a precedent for future work in other domains like RLHF, as acknowledged by reviewers 1 (STEM) and 3 (GyXC).
>
> >  Use of Other Probing Methods (Question 1)
>
> We appreciate your insightful suggestion about incorporating other probing methods. We are indeed fortunate that the type of knowledge we consider enjoys a rich literature of probing methods.
> In the preliminary stage of the work, we considered various methods and chose soft-prompts introduced by Qin et al [2]. This method was selected for its strong performance, especially when applied to smaller models, which tend to exhibit more inconsistencies when interacting with different discrete prompts. Moreover, the simplicity and robustness of this approach have made it well-established in the field, as evidenced by prior work [5,6].
>
> It is important to note that our framework treats the knowledge identification probing mechanism as a black-box component, without making any assumptions about its internal workings. This is because the evaluation protocol makes an observation *only* on a *set of knowledge the probe reliably identifies*. While comparing different probing methods is not the focus of our work, the design of our paradigm allows for easy adaptability; if a more advanced or effective probing method becomes available, it can be seamlessly integrated into the existing framework.
>
> >  Motivation Behind Document Retrieval (Question 3)
>
> Thank you for posing this insightful question.. As outlined in lines 144-151 of our paper, it is crucial that the selected downstream task directly reflects the model's ability to utilize its parametric knowledge. The failure in successfully performing the task should not be attributed to other factors. In document retrieval task, the model is responsible for matching a query to the correct document solely based on its parametric knowledge ($\mathcal{D}^\theta$). If the downstream task included an additional component like reasoning, it would complicate the interpretation of the model's performance, leaving us uncertain whether any failure was due to inadequate reasoning skills or insufficient utilization of pre-trained knowledge. We believe document retrieval as a task not only meets the framework's criteria and enables us to perform systematic study avoid of confounders but also makes our findings even more interesting, as it shows models fail even in this seemingly simple case.
>
> We hope that the clarifications provided have resolved any lingering questions. Would you kindly consider increasing your scores given the main clarifying points above?
>
>
> - [1] Petroni et al, https://aclanthology.org/D19-1250/, EMNLP 2019
> - [2] Qin et al, https://aclanthology.org/2021.naacl-main.410/, NAACL 2021
> - [3] Shin et al, https://aclanthology.org/2020.emnlp-main.346/, EMNLP 2020
> - [4] Newman et al, https://openreview.net/forum?id=DhzIU48OcZh, ICLR 2022
> - [5] Lester et al, https://arxiv.org/abs/2104.08691, 2021,
> - [6] Schucher et al, https://aclanthology.org/2022.acl-short.17.pdf, ACL 2022
> - [7] Hao et al, https://arxiv.org/abs/2206.14268, ACL 2023 Findings

---

### Official Review · Reviewer_STEM · 2023-08-06

**Soundness:** 3

**Excitement:**

3: Ambivalent: It has merits (e.g., it reports state-of-the-art results, the idea is nice), but there are key weaknesses (e.g., it describes incremental work), and it can significantly benefit from another round of revision. However, I won't object to accepting it if my co-reviewers champion it.

**Missing References:**

- The related work section should also cover work highlighting robustness problems of knowledge extraction.

**Paper Topic And Main Contributions:**

The authors introduce a framework to measure parametric knowledge utilisation of PLMs in downstream tasks. Instead of crowd-sourced tasks, they create the tasks from the model’s own knowledge. They first extract knowledge from a PLM (zero-shot tail prediction), subsequently design a task around this knowledge (document retrieval) and fine-tune the model. They target encyclopaedic knowledge. The authors show that i) PLMs did not acquire all factual knowledge par of Wikipedia, ii) not all acquired knowledge is utilised in downstream tasks, iii) model scaling does not close the knowledge utilisation gap and iv) fine-tuning distribution shifts negatively impact knowledge utilisation.

**Questions For The Authors:**

A) A comment about line 043: I would argue that it is commonly assumed that as-is after pre-training an LLM it is often not the case that the acquired knowledge is readily applied to downstream tasks and that different fine-tuning approaches aim to exactly improve that are great examples that demonstrate how more parametric knowledge can be harvest. So, I would not think that it is actually a common assumption that 'if the model poses a certain type of knowledge, it utilises it when performing down-stream tasks.'. Nonetheless, I find it important to quantify and understand when it is used and when it is not.
B) A comment about line 147: You mean ideally for your purpose because I would say generally it is not necessarily the case that we want to ensure that 'failure in performing the task is not due to a lack of pre-training knowledge."
C) Are you ensuring that each test instance has also at least one candidate that has exactly the same head entity combined with exact same relation phrasing as the gold candidate but only differs in terms of the tail entity?
D) I wonder how reliable the identification of knowledge is. One could have also used the retrieval task to identify knowledge and then then tail prediction as the downstream task. I would guess the findings are similar, no? So, I would guess that the zero-shot tail prediction also does not identify all knowledge the model captures and suffers form the same knowledge utilaisation problem.
C) I would love to see a qualitative error analysis.

**Reasons To Accept:**

- The framework to test knowledge utilisation is novel and highlights an interesting research direction. Their framework is useful to track progress across different fine-tuning methods in terms of knowledge utilisation.
- The authors carry out interesting additional analysis. Specifically, their findings around scaling and how it is not reducing the knowledge utilisation gap.

**Reasons To Reject:**

- Their identification of two types of knowledge gaps itself is not novel. We know that large amounts of factual knowledge are not captured  by the PLM (prior papers have shown that performance on zero-shot Q&A or tail entity prediction is not perfect). Already the fact that predictions across paraphrases, different language queries or styles of querying for knowledge (Q&A vs. multiple-choice vs. tail prediction) leads to different predictions, shows that there is a knowledge utilisation gap. Therefore, some of their findings are over-claimed, e.g., "We discover that, [...] PLMs face difficulty utilizing their full knowledge in downstream tasks."
- The method of knowledge identification potentially suffers the knowledge utilisation problem itself (see question D).
- They could have evaluated different fine-tuning paradims. The authors speculate how their finding could translate to RLHF but have not carried out experiments around that.

**Reproducibility:**

5: Could easily reproduce the results.

**Reviewer Confidence:**

3: Pretty sure, but there's a chance I missed something. Although I have a good feel for this area in general, I did not carefully check the paper's details, e.g., the math, experimental design, or novelty.

**Typos Grammar Style And Presentation Improvements:**

- Caption in Figure 5: 'Gap 2 exists in how many of the known facts the model can actually utilize in downstream tasks (the usable knowledge).' There is something wrong with this sentence.

---

> ### Author Rebuttal · Authors · 2023-08-29
>
> Dear Reviewer,
>
> We greatly appreciate your thorough review and valuable feedback. It is heartening to observe that you found our framework **"novel in testing knowledge utilization,"** and that you consider our **analysis to be "interesting."** We're also encouraged by your recognition of the **impact of our work,** notably in charting an "interesting research direction" and in "tracking progress for finetuning methods."
>
> Similarly, we're thankful for Reviewer 3 (GyXC), who commended our **"systematic" and "careful" design** of the framework. This enhances our confidence in the rigor of our approach, and is in alignment with Reviewer 2 (twpn)'s positive remarks about the **"comprehensiveness of experiments."** Furthermore, we're honored by the kind affirmation from Reviewers 2 (twpn) and 3 (GyXC) regarding our work's **contribution**. They highlight its role in fostering a "better understanding of knowledge utilization" and in illuminating "the extent to which pretrained knowledge can be leveraged."
>
> We now address the key questions and concerns in detail below:
>
> > Novelty of Findings
>
> Thank you for bringing attention to the perceived overstatement concerning the knowledge gaps in our evaluation. We recognize the importance of clearly defining the scope and context in which our findings apply. Our intention was to examine the utilization of pretraining knowledge in downstream tasks, which in our work, particularly refers to tasks *a PLM is finetuned on* and present the novelty of our  findings particularly on **Gap 2** (Please refer to the discussion below for more details). We have attempted to articulate this in lines 108-111 and provide an example in lines 54-59. However, acknowledging your observation, we realize that further clarification may be necessary to prevent any confusion and plan to revise the introduction and abstract to address this.
>
> In lights of the fruitful discourse started by the reviewer, we would like to clarify some of our points in more details to better ground our discussion:
>
> **Gap 1**, as we mention in the paper, exists in the acquired knowledge from the pretraining stage: Some facts might not have been seen by the model or captured appropriately, suggesting that this gap might be rooted in the model's capacity or the quality of pretraining data. For instance, as depicted in Fig. 1, larger models within the same family generally exhibit more identifiable facts (despite the same pretraining data); LLaMa, with a more expansive training corpus [1], showcases greater identifiable knowledge in comparison to OPT in the same parameter regime. As correctly highlighted by the reviewer, previous research has addressed this gap across various contexts[2,3,4], which we cite in lines 515-524.
>
> **Robustness in Knowledge Identification:** We argue that the robustness in knowledge identification is inherently a property of the knowledge probing method: A PLM might possess knowledge on a subject, but the probe might not detect it (e.g. there are better or worse prompts for extracting some facts). Indeed, Petroni et al [2] acknowledge “...we are measuring a lower bound for what language models know...”. This is why improving the accuracy and reliability of such probes is an active area of research [5,6]. We believe that while the question of this type of robustness is important, it is more concerned with the probing method rather than the model itself and is not the primary focus of our study. Nevertheless, we utilise a standard probing method from existing literature [3], which is known for improved robustness. Should a better probing method emerge, our framework can readily harness it to produce more accurate measurements.
>
> **Gap 2:** We, as discussed in the paper, refer to this gap as potential inefficiency of a PLM in utilizing its knowledge during the finetuning phase for a downstream task. This definition is irrespective of the probing method employed and the probe's capacity to identify the model's knowledge. Our framework employs a specific probing method without making assumptions about its accuracy. In doing so, it measures the utilization gap solely on the *knowledge set correctly identified by the probing method*, circumventing the imperfections of the probe. We believe findings of this framework points to an aspect of model behavior that remains relatively unexplored in current literature. For example, Gudibande et al [7] recently attributed the underwhelming performance of open-source LLMs (in comparison to OpenAI models) to a better base pretraining than to the finetuning or the instruction data. Our framework's meticulous design, recognized by reviewer GyCX, allows us, to the best of our knowledge, to demonstrate and accurately quantify this gap for the first time. Nonetheless, we are more than happy to explore any references the reviewer might be aware of that address and quantify this specific gap.
>
> We understand that these details may not have been sufficiently discussed in the original draft but we make sure to have a more comprehensive discussion in the updated manuscript.
>
> > Reliability of Knowledge Identification & Question D
>
> Thank you for raising the concern regarding the reliability and robustness of knowledge identification probing. As mentioned in the previous section (*Robustness in Know....*), we argue that the robustness of knowledge identification is predominantly determined by the probing technique rather than the intrinsic properties of the model itself. We differentiate this from the knowledge utilization issue happening in the model (after finetuing) that we are exploring in gap 2. Additionally, we would like to note that our framework takes such innate unreliability into consideration by only quantifying the gap 2 on *the set of knowledge the probe can correctly and reliably identify*. Thus, we expect a consistent conclusion with a different probing method as our framework only makes observations about the identified knowledge.
>
> We appreciate the reviewer’s comment about using “retrieval” in knowledge probing. Indeed, this paradigm is agnostic to the type of tasks employed at various stages. Employing "retrieval" as knowledge identification, however, might be challenging since this task is vastly different from what a PLM is usually trained on (e.g. language modeling). The model could potentially underperform in a zero-shot retrieval setting, leading to a suboptimal probing result.
>
> > The Prevalence of Knowledge Utilization Assumption in the Literature (Question A)
>
> We acknowledge the reviewer's sentiment about the prevalence of such assumptions in the literature. To address this, we’d like to highlight some quotes from the literature that explicitly make this assumption: Zhou et al. 2020 [8] states that “syntactic, semantic and word sense knowledge are contained in such [PLM] representations, which explains why they benefit such tasks.” Furthermore, a growing body of recent work proposes to use minimal instruction-tuning datasets [9,10,11], (some as limited as 1000 examples), motivated by the hypothesis that instruction-tuning only teaches the model to interact with the user and utilizes the pretraining knowledge. Particularly, Zhou et al, 2023 [9] states that “alignment can be a simple process where the model learns ...[the] format for interacting with users, to expose the knowledge [...] already acquired during pretraining”. We agree with the reviewer that such assumptions might have become less dominant, but we hope our work sheds light on the hidden consequences of them, especially given the emergence of such recent works.
>
> > Question B
>
> Thank you for your comment. Yes, we indeed meant “ideally” with respect to the goal of our evaluation. We’ll revise the wording to avoid any misinterpretation.
>
> > Question C
>
> Yes, candidates with the same head entity and relation phrasing as the gold standard, but different tail entities, are included. This type of candidate is the one we care the most, and we’ve ensured it’s well represented in the task construction. We tried to convey this in lines 264-275. We will improve the clarity in this section.
>
> > Additional Qualitative Error Analysis
>
> We appreciate the reviewer's interest in a qualitative error analysis of models' performance. In our preliminary analysis, we focused on evaluating the distribution of similarity scores $sim(q, .)$ (line 227) between the query and different candidate types. Our findings indicate that the model faces more difficulties in distinguishing among candidate types $(\mathbf{h}, \mathbf{r}, \cdot)$, $(\mathbf{h},, \cdot, \mathbf{t})$, and $(\cdot,, \mathbf{r}, \mathbf{t})$. Specifically, these candidate types generate similarity scores that exhibit overlapping distributions with those of the gold documents.
>
> This issue could potentially be attributed to the nature of these candidate types, where only one entity element—be it the head, relation, or tail—is distinct from the query, thereby confounding the model's decision-making process. Additionally, we discovered an intriguing behavior in our models: they struggle to differentiate between unfactual and factual candidate types both denoted as $(\cdot, \cdot, \cdot)$. We hypothesize that this lack of distinction could be a consequence of these candidates sharing no entities with the query, leading the model to deprioritize or even discard them.
>
> We find these patterns to be consistent across the models (BERT, RoBERTa, GPT-Neo, and OPT) and will include this analysis in the appendix for further scrutiny.
>
>
> > Application to RLHF domain
>
> The reviewer makes an astute observation regarding the potential applicability of the framework to the RLHF domain. This connection, as also noted by reviewer GyxC, is indeed intriguing and noteworthy. While our proposed paradigm is compatible with instruction-tuning, there are several critical aspects that must be meticulously designed to accurately isolate the phenomenon of knowledge utilization. Specifically, the development of a method for 'task' extraction from scratch, for the knowledge identification phase, is non-trivial as there is no well-established technique available in the literature. Moreover, the process of converting the extracted knowledge into a downstream task remains ambiguous and needs exploration. Given these complexities, we believe that a comprehensive analysis of both knowledge types surpasses the scope of a single paper. Nevertheless, we recognize the intriguing nature of this question and believe that the findings in our work could motivate such in-depth future studies, opening up an important avenue for research.
>
> We hope the clarifications provided have resolved any remaining questions. Would you kindly consider increasing your ratings based on the key points clarified?
>
>
> - [1] Touvron et al. https://arxiv.org/abs/2302.13971, 2023
> - [2] Petroni et al, https://aclanthology.org/D19-1250/, EMNLP 2019
> - [3] Qin et al, https://aclanthology.org/2021.naacl-main.410/, NAACL 2021
> - [4] Tenney et al, https://openreview.net/forum?id=SJzSgnRcKX, ICLR 2019
> - [5] Shin et al, https://aclanthology.org/2020.emnlp-main.346/, EMNLP 2020
> - [6] Newman et al, https://openreview.net/forum?id=DhzIU48OcZh, ICLR 2022
> - [7] Gudibande et al, https://arxiv.org/abs/2305.15717, 2023
> - [8] Zhou et al, https://arxiv.org/abs/1911.11931, AAAI 2020
> - [9] Zhou et al, https://arxiv.org/abs/2305.11206, 2023
> - [10] Touvron et al, https://arxiv.org/abs/2307.09288, 2023
> - [11] Taori et al, https://github.com/tatsu-lab/stanford_alpaca, 2023

---

### Meta-Review · Area_Chair_XiqM · 2023-09-17

**Recommendation:** 4

**Metareview:**

The reviewers agree this work conducts a comprehensive investigation into an important topic of analysis, and yields insights into the changing nature of the knowledge acquisition and utilization gaps with model scale. While reviewers note challenges in deriving robust and generalizable findings without exploring many datasets, finetuning and probing methods, they do acknowledge the authors have thought deeply about the problem and experimental setup to make their findings as useful and possible.

---

### Decision · Program_Chairs · 2023-10-07

**Decision:**

Accept-Findings

**Comment:**

The reviewers agree this work conducts a comprehensive investigation into an important topic of analysis, and yields insights into the changing nature of the knowledge acquisition and utilization gaps with model scale. While reviewers note challenges in deriving robust and generalizable findings without exploring many datasets, finetuning and probing methods, they do acknowledge the authors have thought deeply about the problem and experimental setup to make their findings as useful and possible.